# The Analysis of the Expected Change in the Classification Probability of the Predicted Label

**Ruo Yang**                                                                      *ryang23@hawk.iit.edu*
*Department of Computer Science, Illinois Institute of Technology*
*Chicago, IL, USA*

**Ping Liu**                                                                       *piliu@linkedin.com*
*LinkedIn Corporation*
*Mountain View, CA, USA*

**Mustafa Bilgic**                                                                 *mbilgic@iit.edu*
*Department of Computer Science, Illinois Institute of Technology*
*Chicago, IL, USA*

**Reviewed on OpenReview:** *https://openreview.net/forum?id=gvqzvUVPiQ*

## Abstract

We present a formalism for estimating the expected change in the probability distribution of the predicted label of an object, with respect to all small perturbations to the object. We first derive analytically an estimate of the expected probability change as a function of the input noise. We then conduct three empirical studies: in the first study, experimental results on image classification show that the proposed measure can be used to distinguish the not-robust label predictions from those that are robust, even when they are all predicted with high confidence. The second study shows that the proposed robustness measure is almost always higher for the predictions on the corrupted images, compared to the predictions on the original versions of them. The final study shows that the proposed measure is lower for models when they are trained using adversarial training approaches.

## 1 Introduction

Deep learning models are used for numerous industrial, governmental, and personal applications that include medical image analysis Liu & Bilgic (2021), machine translation Sutskever et al. (2014), face recognition Schroff et al. (2015), automated driving Chen et al. (2017), loan application evaluations Petropoulos et al. (2019), and recommender systems Cheng et al. (2016). As deep learning models have been deployed for critical applications such as automated driving and medical diagnosis and as it has been found that these models are vulnerable to simple noise and adversarial attacks Szegedy et al. (2014), the analysis of the robustness of these models has been of paramount importance Athalye et al. (2018).

The robustness of the model's prediction *probability* is crucial for decision making, including how much confidence to associate with that prediction, which action to take, and what information to gather next. For example, in medical diagnosis, the predicted diagnosis is rarely useful by itself; the classifier's confidence in that prediction as well as the robustness of that prediction are crucial for making treatment plans and ordering additional diagnostic tests.

It is known by the research community that even high probability predictions of a highly accurate model might be unreliable Goodfellow et al. (2015). For example, if a model is a high-variance model or if its probabilities are not calibrated, its predictions can be unstable and its probability outputs cannot be used to indicate uncertainty Platt et al. (1999). Even for models whose probability distributions are well calibrated, if the object under consideration lies on a region where the probability distribution changes drastically,

the probability prediction might be unstable. Object $x_2$ in Figure 1 demonstrates this situation where the probability distribution has high curvature around the input, and the predicted probability distributions of $x_2$'s close neighbors are different from the probability value of $x_2$.

In this paper, we study, analytically and empirically, quantifying the predicted probability difference between an object $x$ and all hypothetical objects at $\Delta x$ distance to $x$, i.e., $x + \Delta x$ for all small $\forall \Delta x \in \mathbb{R}^n$. We derive an estimate of this measure through Taylor expansion and the Divergence theorem. We then conduct empirical analyses on four datasets. Note that one standard strategy to estimate this measure is to sample $\Delta x$. While an $x$ has only two neighbors in 1D, the number of $\Delta x$ is infinite even in 2D; hence, we provide an analytical derivation, rather than reverting to sampling. Our main contributions include:

- We prove that the expected change in the probability distribution of the object, with respect to *all* small perturbations around the object, is proportional to the Laplace operator ($\mathbb{LO}$), which is defined as the sum of the second partial derivative of the classifier's output with respect to each input dimension[1].

- We conduct three empirical studies on four datasets, analyzing the derived estimate ($\mathbb{LO}$) and its relationship to the robustness of the predicted label to random noise, the robustness of the prediction to several types of image corruption (blur, brightness, etc.), and comparing the robustness of models when they are trained using traditional training methods versus adversarial training approaches.

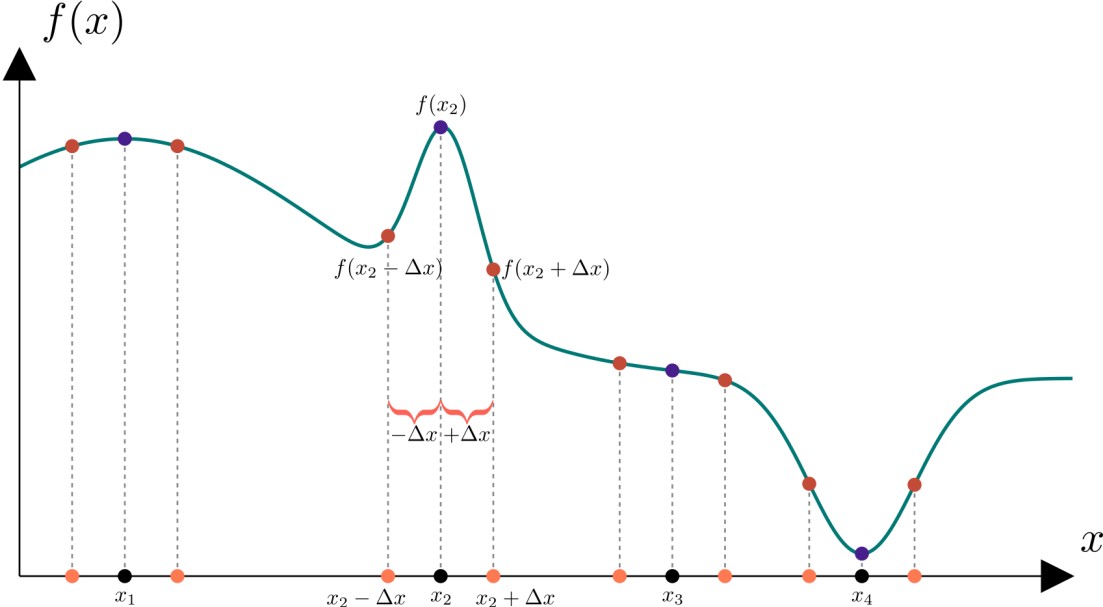

Figure 1: Illustration of the expected probability difference between an object $x$ and its close neighbors $x + \Delta x$, i.e., $\mathbb{E}_{\Delta x}(f(x + \Delta x) - f(x)) = \mathbb{E}_{\Delta x}(\Delta f(x))$. In this simple 1 feature classifier, the expected value is computed with its left and right neighbours. In this example, $\mathbb{E}_{\Delta x}(\Delta f(x_1)) < 0$, $\mathbb{E}_{\Delta x}(\Delta f(x_2)) \ll 0$, $\mathbb{E}_{\Delta x}(\Delta f(x_3)) \approx 0$, and $\mathbb{E}_{\Delta x}(\Delta f(x_4)) \gg 0$. We provide analytical and empirical analyses of $\mathbb{E}_{\Delta x}(\Delta f(x))$ for all small $\forall \Delta x \in \mathbb{R}^n$.

The rest of this paper is organized as follows. We discuss related work in Section 2. We formulate the problem and derive its analytical estimate in Section 3. We present our empirical analyses in Section 4. We then discuss the limitations and future directions in Section 5 and conclude in Section 6.

---

[1]The exact constant is provided later in Section 3.

## 2 Related Work

Studies show that the state-of-the-art deep neural network models are vulnerable to adversarial examples that are created with small perturbations to the original examples Szegedy et al. (2014). For example, Goodfellow et al. (2015) showed that two images that are indistinguishable to humans could mislead high-accuracy neural networks. Several papers focused on constructing adversarial examples Kurakin et al. (2017); Xiao et al. (2018), measuring the overall robustness of the models to attacks Peck et al. (2017), and training the models to be more robust to attacks Madry et al. (2018); Jakubovitz & Giryes (2018). While the robustness of the models to adversarial attacks is a closely related area, our setting is not the adversarial setting. The objective of this paper is to evaluate the robustness of the predicted probability against random noise instead of adversarial attacks. We focus on quantifying the expected change to the probability distributions under small and random perturbations to the data.

Another related area of work is the calibration and quantification of the uncertainty of the predictions of neural networks. Even though the outputs of the neurons passed through a sigmoid (or a softmax) function are often treated as a probability distribution, these distributions can often be at the extreme ends of $[0, 1]$, displaying an "over-confidence" that is not necessarily warranted by the data. Hence, several papers studied calibrating the probability distributions of these models Guo et al. (2017); Corbière et al. (2019); Jiang et al. (2018); Lakshminarayanan et al. (2017). Alternatively, the Bayesian modeling and inference approaches for neural networks Denker & LeCun (1990); MacKay (1992) compute a distribution over the predicted probability distribution and hence provide a mechanism to measure the uncertainty of the prediction probability itself. Finally, several approaches used alternative mechanisms for quantifying the uncertainty of the neural networks. For example, while the drop-out Srivastava et al. (2014) is often used to prevent overfitting, it can also be used for measuring the uncertainty of the model Gal & Ghahramani (2016).

A closely related work to ours is the work by Jiang et al. (2018). They estimate the uncertainty of the predicted label of an object as the ratio between the distance to the closest neighbor that has the same predicted label and the distance to the closest neighbor that has a different predicted label. We estimate the expected change in the probability distribution of the object's label with respect to *all* possible small perturbations to the object, whereas Jiang et al. (2018) estimate the uncertainty of an object's label via the nearest two objects in its neighborhood.

Our focus in this paper is orthogonal to the approaches that quantify and calibrate the uncertainty of neural networks. Regardless of whether the probability distribution is calibrated or not, and regardless whether the method is Bayesian or based on drop-out, we quantify how much the predicted probability distribution is expected to change with respect to small perturbations to an object. Even though this is similar to the Bayesian approach where a confidence around the probability prediction can be computed, the Bayesian approach is with respect to the posterior distribution, which is a combination of the prior distribution and the observed data. Our approach, on the other hand, measures the robustness of the probability distribution with respect to small perturbations to an object, measuring robustness to noise, and identifying probability regions where the distribution is expected to stay the same or change drastically.

## 3 Our Approach

Let $x \in \mathbb{R}^n$ be a point of interest and $F(x) : \mathbb{R}^n \to [0, 1]^m$ be a differentiable probabilistic classifier where the number of classes is $m$. Let $F_c(x)$ be the probability that $x$ belongs to class $c$, $1 \leq c \leq m$. We then denote $f(x) = \max_{1 \leq c \leq m} F_c(x)$ as the probability of the predicted class. Let $S$ be a $n$-dimensional sphere centered at $0$ with radius $r \geq 0$. Consider a random perturbation vector $\Delta x$ that is uniformly distributed on the sphere $S$, i.e., $\Delta x \sim \text{Unif}(S)$. Then $x + \Delta x$ represents a displacement around $x$ with a random noise of length $r$. Further assume that $V$ is an $n$-dimensional ball centered at $0$ with radius $r$, such that $S$ is the surface of the ball $V$, i.e. $\partial V = S$.

Our goal is to quantitatively measure the vulnerability of the classifier's predicted probability on the point $x$ for a class $c$ against random input perturbations. In other words, we are interested in computing the expected change in the probability of the predicted class $c$ with respect to *all* possible random perturbations $x + \Delta x$:

$$\mathbb{E}_{\Delta x}(\Delta f(x)) \equiv \mathbb{E}_{\Delta x}(f(x + \Delta x) - f(x)). \tag{1}$$

One way to approach this problem is to use Monte Carlo sampling for perturbation $\Delta x$, then take an expectation of the probability change with respect to the sampled perturbations. While sampling is a viable approach, it comes with its own challenges, such as the need for generating large number of samples to create reliable estimates, and more importantly, it requires the noise level as a hyper-parameter. We tackle the problem of determining this quantity via analytical methods, without resorting to sampling, and solve it as a function of the noise level. We first present the main theorem of our paper and then describe the general sketch of the proof. We provide the detailed steps of the proofs in the supplementary material.

**Theorem 1.** *Denote $\nabla^2$ as the Laplace operator, and $\mathbb{LO}$ as the Laplacian of a given function $g : \mathbb{R}^n \to \mathbb{R}$ at point $x$. That is, $\mathbb{LO} = \nabla^2 g(x) = \sum_{i=1}^{n} \frac{\partial^2 g(x)}{\partial x_i^2}$. Then, the expected change in the probability distribution of the predicted class of $x$, with respect to all random perturbations $\Delta x$ is:*

$$\mathbb{E}_{\Delta x}(\Delta f(x)) = \frac{r^2}{2n} \nabla^2 f(x) + O(|\Delta x|^3) \simeq \frac{r^2}{2n} \mathbb{LO}. \tag{2}$$

As defined earlier, all possible perturbations $\Delta x$ with $L_2$ norm $r$ uniformly live on the surface of $S$. This indicates that the probability density function of the perturbation, $p(\Delta x)$, is a constant with respect to perturbations. Furthermore, the expected change in the probability of the predicted class $c$ is the integration of the probability variation caused by the perturbation $\Delta x$ in expectation to its corresponding probability density function. Formally:

**Lemma 2.** *Denote $\Gamma(.)$ as the Gamma Function, then the expected change in the value of $f$ around input $x$ is:*

$$\mathbb{E}_{\Delta x}(\Delta f(x)) = \frac{\Gamma(\frac{n}{2})}{2\pi^{\frac{n}{2}}} r^{1-n} \int_S \Delta f(x) dS. \tag{3}$$

We prove Lemma 2 in the supplementary material. We next use the second-order Taylor expansion for estimating $f(x + \Delta x)$:

$$f(x + \Delta x) = f(x) + \sum_{i=1}^{n} \frac{\partial f(x)}{\partial x_i} \Delta x_i + \frac{1}{2} \sum_{i=1}^{n} \frac{\partial^2 f(x)}{\partial x_i^2} \Delta x_i^2 + \frac{1}{2} \sum_{i=1}^{n} \sum_{i \neq j}^{n} \frac{\partial^2 f(x)}{\partial x_i \partial x_j} \Delta x_i \Delta x_j + O(|\Delta x|^3). \tag{4}$$

Then, the integral of $\Delta f(x)$ over all possible random perturbation $\Delta x$ can be estimated as:

$$\int_S \Delta f(x) dS \simeq \int_S (\sum_{i=1}^{n} \frac{\partial f(x)}{\partial x_i} \Delta x_i + \frac{1}{2} \sum_{i=1}^{n} \frac{\partial^2 f(x)}{\partial x_i^2} \Delta x_i^2 + \frac{1}{2} \sum_{i=1}^{n} \sum_{i \neq j}^{n} \frac{\partial^2 f(x)}{\partial x_i \partial x_j} \Delta x_i \Delta x_j) dS. \tag{5}$$

The Divergence theorem states that the volume integral of a vector field over a close region inside of a surface equals the surface integral of the same vector field over the close surface. For the ball $V$, the unit normal vector directed outward from $V$ is $\overrightarrow{n} = (\frac{\Delta x_1}{r}, \frac{\Delta x_2}{r}, \ldots, \frac{\Delta x_n}{r})$.

**Lemma 3.** *We have $\int_S \Delta x_i dS = 0$, $\int_S \Delta x_i^2 dS = \frac{\pi^{\frac{n}{2}}}{\Gamma(\frac{n}{2}+1)} r^{n+1}$, and $\int_S \Delta x_i \Delta x_j dS = 0$, for $1 \leqslant i, j \leqslant n$ with $i \neq j$.*

In Equation (5), the $\frac{\partial f(x)}{\partial x_i}$, $\frac{\partial^2 f(x)}{\partial x_i \partial x_j}$, and $\frac{\partial^2 f(x)}{\partial x_i^2}$ are independent constants from the variable $\Delta x_i$. By Lemma 3 we have:

$$\int_S \sum_{i=1}^n \frac{\partial f(x)}{\partial x_i} \Delta x_i dS = 0, \text{ and } \int_S \frac{1}{2} \sum_{i=1}^n \sum_{i \neq j}^n \frac{\partial^2 f(x)}{\partial x_i \partial x_j} \Delta x_i \Delta x_j dS = 0.$$

Using the lemmas above, we derive that the expected change in the probability of the predicted class with respect to all possible random perturbations with length $r$ has the following approximate value:

$$\mathbb{E}_{\Delta x}(\Delta f(x)) \simeq \frac{\Gamma(\frac{n}{2})}{2\pi^{\frac{n}{2}}} r^{1-n} \left(\frac{1}{2} \sum_{i=1}^n \frac{\partial^2 f}{\partial x_i^2} \frac{\pi^{\frac{n}{2}}}{\Gamma(\frac{n}{2}+1)} r^{n+1}\right)$$

$$= \frac{r^2}{4} \frac{\Gamma(\frac{n}{2})}{\Gamma(\frac{n}{2}+1)} \sum_{i=1}^n \frac{\partial^2 f}{\partial x_i^2} = \frac{r^2}{2n} \nabla^2 f(x). \tag{6}$$

concluding the proof of Theorem 1.

Given an $n$ dimensional input $x$ and a fixed radius $r$, Theorem 1 shows that the expected change in the probability of the predicted class $c$ at $x$ with respect to all possible permutations is linearly proportional to $\mathbb{LO}$. If the $\mathbb{LO}$ is a large negative (positive), we expect that the predicted probability at $x$ decreases (increases) rapidly due to random perturbations. On the other hand, if $|\mathbb{LO}|$ is small and close to zero, Theorem 1 suggests that the predicted probability is robust to noise at the input.

### 3.1 $\mathbb{LO}$ in Practice

There are three tasks where the quantification of the expected change in the predicted probability distribution can be used in practice. We first discuss the robustness of the predicted label of a classifier, then the robustness of the predicted probability distribution, and finally the robustness of the model.

#### 3.1.1 The Robustness of the Predicted Label

In the first task, we desire to know if the predicted label by the classifier is robust under noise. One way to measure the robustness of the predicted label to noise is to sample white noise, apply it to the test object, and analyze number of times the predicted label changes.

Another approach for measuring the robustness of the predicted label to noise is to calculate the uncertainty of the prediction. If the object is close to the decision boundary, then its label is not expected to be robust to noise. The distance to the decision boundary can be calculated in several ways. For example, for support vector machines, the distance to the margin can be used. For probabilistic classifiers, the probability output of the classifier can be used. In the binary classification case, if the prediction probability is close to 0.5, then the object is treated as uncertain. In the multi-class case, one can use $1 - p_c$ where $p_c$ is the probability of the predicted class, or the margin $p_c - p_n$ where $p_c$ is the probability of the predicted class and $p_n$ is the probability of the next likely class, or the entropy of the distribution $-\sum p_i \log p_i$.

We hypothesize that the value of $\mathbb{LO}$ can be used to identify objects whose label prediction is not uncertain and yet the prediction is still not robust to noise. Let us analyze the binary classification and multiclass classification cases separately. Let $\mathbb{LO}(i)$ be the $\mathbb{LO}$ value for class $i$. In the binary case, let the classes be $A$, $B$. In this case, we have $\mathbb{LO}(A) = -\mathbb{LO}(B)$. Without loss of generality, let the predicted class be $A$ and let its probability be $p_A$. If $\mathbb{LO}(A) \ll 0$, then the probability of class $A$ is expected to drop in expectation with respect to perturbations and hence the predicted label is not expected to be robust. On the other hand, if $\mathbb{LO}(A) \approx 0$ or better yet if $\mathbb{LO}(A) > 0$, the prediction of label $A$ is expected to be robust to noise. In the multiclass case, assume there are $m$ classes, let the predicted class be $c_i$ and its probability be $p_{c_i}$. If $\mathbb{LO}(c_i) \approx 0$ or better yet if $\mathbb{LO}(c_i) > 0$, the prediction of label $c$ is expected to be robust to noise. If $\mathbb{LO}(c_i) \ll 0$, however, then the prediction is not expected to be robust. Further, the summation of $\mathbb{LO}$ cross all classes is 0, e.g. $\sum_{i=1}^m \mathbb{LO}(c_i) = 0$. When $\mathbb{LO}(c_i) \ll 0$, observing $\mathbb{LO}(c_j) \gg 0$ for some $i \neq j$ adds further evidence to the unstability of the predicted label $c_i$ to noise.

### 3.1.2 The Robustness of the Probability Distribution

In this task, rather than the predicted label, we are interested in the robustness of the probability distribution itself to small perturbations. Theorem 1 states that the expected probability change is approximately equal to $\frac{r^2}{2n}\mathbb{LO}$. We emphasize that $\frac{r^2}{2n}\mathbb{LO}$ approximates the expectation of the change, which is the *mean* change, and it is *not* an estimate of the *variance*. Hence, $\mathbb{LO} \approx 0$ means the mean change of the probability distribution with respect to the perturbations to an object is close to 0 and hence the probability in the object's neighborhood is approximately equal to the object's probability on *average*. Therefore, when $\mathbb{LO} > 0$ or $\mathbb{LO} < 0$, we can claim that the predicted probability is not robust to perturbation, whereas when $\mathbb{LO} \approx 0$, we make no claims regarding the robustness of the probability distribution. All we can claim when $\mathbb{LO} \approx 0$ is that the object's probability distribution is similar, on average, to the distribution in its immediate neighborhood within $r$.

Theorem 1 measures $\mathbb{E}_{\Delta x}(\Delta f(x))$, which is useful for identifying objects that have high confidence but unstable predictions. Another measure that would be useful to determine the robustness of the probability distribution is $\mathbb{E}_{\Delta x}(|\Delta f(x)|)$. Using triangle inequality and the theorems from the prior work Sỳkora (1974), we prove in the supplementary material the following bound:

$$
\begin{aligned}
\mathbb{E}_{\Delta x}(|\Delta f(x)|) &\simeq \frac{\Gamma(\frac{n}{2})}{2\pi^{\frac{n}{2}}} r^{1-n} \int_S |\Delta f(x)| dS \\
&\leq \frac{r}{\sqrt{\pi}} \frac{\Gamma(\frac{n}{2})}{\Gamma(\frac{1}{2}+\frac{n}{2})} \sum_{i=1}^n |\frac{\partial f(x)}{\partial x_i}| + \frac{r^2}{n\pi^2} \sum_{i=1}^n \sum_{i\neq j}^n |\frac{\partial^2 f(x)}{\partial x_i \partial x_j}| + \frac{r^2}{2n} \nabla^2 f(x).
\end{aligned} \tag{7}
$$

### 3.1.3 The Robustness of the Model

In this task, we compare different models in terms of their robustness to noise. A robust model's predicted probability distribution is expected to change less compared to a less robust one, under minor changes to the input. In essence, this is also related to the variance of the models. Given two models that are comparably accurate on a validation set, the model that is more robust to noise is preferred as it is expected to generalize better to unseen data.

We illustrate an extreme case of this phenomenon in Figure 2. The more robust model has smooth curvature in its probability distribution, and hence the noise in inputs causes smaller changes in the predicted probability values when compared to the non-robust model. Because $\mathbb{LO}$ is proportional to the expected change in the probability distribution with respect to all perturbations per Theorem 1, we hypothesize that the absolute value of $\mathbb{LO}$ can be used to compare the robustness of two models. More specifically, we propose computing the expected $|\mathbb{LO}|$ for a given model $A$:

$$
\int_x p(x) |\mathbb{LO}_A(x)|
$$

In the experiments section, we estimate this measure using a held-out set, $T$:

$$
\frac{1}{|T|} \sum_{x \in T} |\mathbb{LO}_A(x)|
$$

Given two models $A$ and $B$, the model that has the higher value of this measure is expected to be less robust to noise. We present experimental results in Section 4.4, using this measure to compare models that are trained "normally" versus "adversarial training methods."

## 4 Experimental Methodology and Results

We conduct three empirical studies. In the first study, we study the relationship between $\mathbb{LO}$ and the stability of the label prediction on four image classification datasets. Second, we compare the magnitude of $|\mathbb{LO}|$ on the original versus corrupted versions of images on two benchmark datasets. Finally, we compare $\mathbb{LO}$ on models that are trained using traditional approaches versus adversarial training methods.

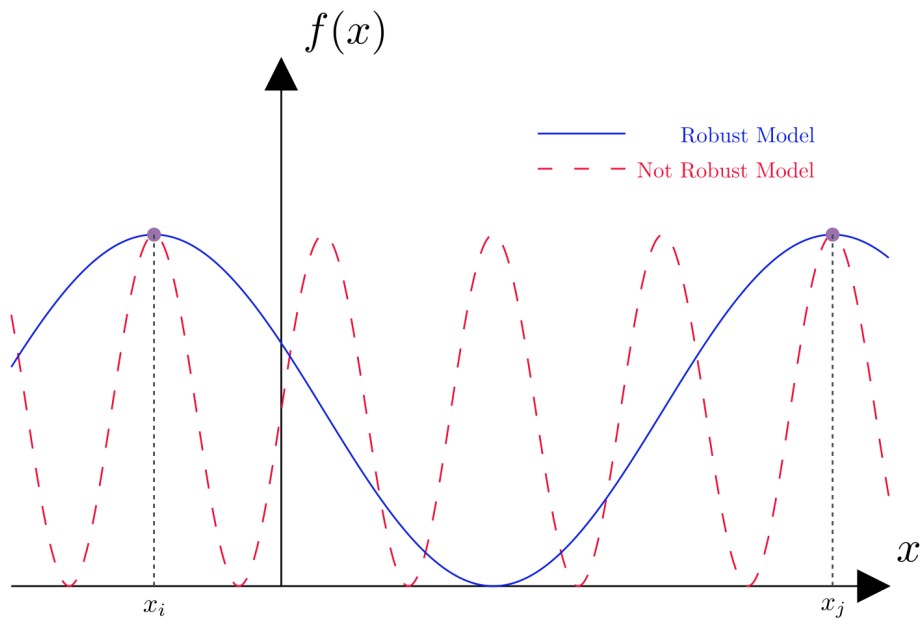

Figure 2: The illustration of the robustness of two models in 1D. $x_i$ and $x_j$ are two example test instances. Both robust model (blue solid line) and not-robust model (red dashed line) have the same prediction for instances $x_i$ and $x_j$. However, the curvatures of the robust model are smoother at those two points. In this case, $|\mathbb{LO}|$ computed using the robust model is lower than the one computed using the not-robust model.

### 4.1 Datasets and the Models

We conduct experiments on the MNIST digit classification LeCun et al. (1998), CIFAR-10 object classification Krizhevsky et al. (2009), histology images of colorectal cancer (CRC) dataset Sirinukunwattana et al. (2016), and the street view house numbers (SVHN) dataset Netzer et al. (2011). Theorem 1 applies to any differentiable classifier. We focus on neural networks in this study. We train LeNet LeCun et al. (1998) for MNIST and ResNet He et al. (2016) for the other three datasets. The details of training, validation, and test splits, the model structure, activation functions, and the model hyper-parameters are as follow.

We split the MNIST data as 60K for training and 10K for testing. We adopt the LeNet architecture, use the hyperbolic tangent function for all convolution layers, and linear activations for the dense layers. We use all samples in the training set to train with 20 epochs, batch size of 500, and Adam Kingma & Ba (2015) optimizer with a learning rate of 0.01. The accuracy of this model on the test set is 98%.

The CIFAR-10 dataset consists of 45K training, 5K validation, and 10K testing images Abadi et al. (2015). We create a ResNet model with 3 Res-blocks. We initialize all weights following He et al. (2015a) and utilize the PRelu He et al. (2015b) activation function. We augment the training by flipping the images horizontally and shifting both height and width with a maximum 12.5% range. We use a batch size of 128, regularization constant of 0.0001, learning rate of 0.001, and SGD with the momentum of 0.9. We optimize the epoch number using the validation accuracy. The final model achieves top-1 test accuracy of 85%. We adapt the same model structure of CIFAR-10 to the SVHN dataset. We keep the original test set (26,032 objects) of SVHN for testing, and reserve random 5K objects from training set as the validation set. The final model has a test accuracy of 92%.

We also use the histology images colorectal cancer dataset (CRC), which contains 100 H&E stained colorectal adenocarcinomas images where each image contains several cells. The cells in the stained images are labeled as: *Epithelial*, *Inflammatory*, *Fibroblast*, or *Miscellaneous*, and the location of the center of each labeled cell

is provided in the data. We extract a 27x27x3 image for each cell at the locations provided in the data. The total samples for each class are 7,057, 6,278, 5,130, and 1,842 respectively. We split the dataset into 70% train, 15% validation, and 15% test. We used the same ResNet architecture of SVHN, but changed the input and output size as needed. The model achieves a top-1 test accuracy of 74%.

We implemented both LeNet and ResNet models using Tensorflow Abadi et al. (2015). To compute $\mathbb{LO}$ for each image, we first calculate the Hessian matrix, $H$, of the maximum predicted probability of the image by using the automatic differentiation function provided by Tensorflow. $\mathbb{LO}$ is then simply the trace of the $H$. We used a single modern GPU (Quadro RTX 5000) for the experiments.

## 4.2 $\mathbb{LO}$ and Label Flip

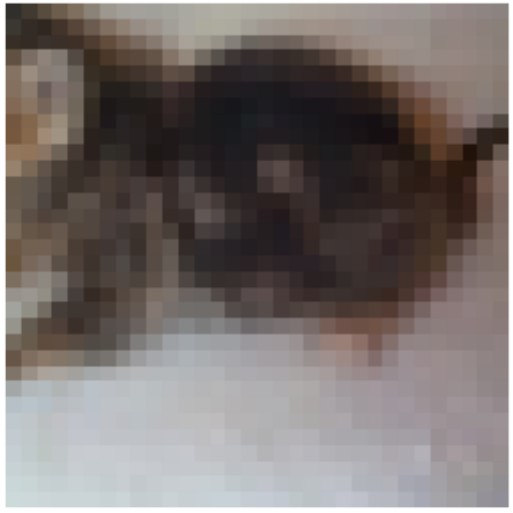 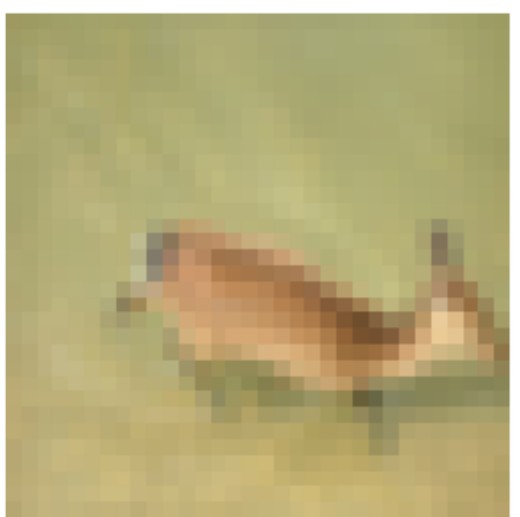



(a) Predicted label: Cat
Prediction prob: 0.803
True label: Cat
$E(\Delta Prob) = -0.230$
$\mathbb{LO} = -1413$

(b) Predicted label: Bird
Prediction prob: 0.814
True label: Deer
$E(\Delta Prob) = -0.248$
$\mathbb{LO} = -1523$



Figure 3: Two example images from the CIFAR-10 dataset. Even though the predictions by the ResNet model are confident, our estimate shows that the prediction probabilities are expected to decrease if small perturbations are applied to these images. Sampling and applying a noise vector of length $r = 1$, which is equivalent to an average 0.037 change per pixel, shows that the cat prediction flips to a different label in 57% of the samples (43% 'cat', 29% 'dog', and 28% 'frog') and the bird prediction flips to a different label in 64% of the samples (36% 'bird', 63% 'deer', and 1% 'frog).

In this section, we study if $\mathbb{LO}$ can be used to identify objects whose label predictions are not robust to noise. As discussed in Section 3.1, even if an object's label is predicted with high confidence, if $\mathbb{LO} \ll 0$ for the predicted class, the prediction is likely to be not robust to noise.

To test our hypothesis, we take a random sample of 1K objects from the test set for each dataset, and we calculate the probability and $\mathbb{LO}$ for their predicted labels. Then, we generate 10K noisy versions of each of these images by sampling a noise vector of length $r$ (we tested r=0.5, 1, and 2) and appending it to the original image. We then compare the predicted labels for the noisy versions of the original images with the original labels.

In the first experiment, we focus on objects where the model is confident in its prediction; we specifically focus on objects whose predicted label has 0.8 or a higher probability value. This filtering results in 903 test objects for MNIST, 891 test objects for CIFAR-10, 603 test objects for CRC, and 805 test objects for SVHN. We group these objects into two: top 50 objects where $\mathbb{LO}$ is lowest (referred as "Bottom") and

top 50 objects where $\mathbb{LO}$ is highest (referred as "Top "). We compute, for each test object, the ratio of its noisy versions that have a different predicted label from the original image (also referred as "label flip"). We report the average label flip rates for both the "Bottom" and the "Top " group, under varying values of $r$, in Table 1. The results show that the label flip percentage is almost always small for the "Top $\mathbb{LO}$" group and the label flip percentage is drastically higher for the "Bottom $\mathbb{LO}$" group. These results indicate that large negative values of $\mathbb{LO}$ ("Bottom") can identify objects whose label predictions are not stable, even when they are predicted with high confidence. We present two example images from the CIFAR-10 dataset in Figure 3. Both images were predicted with high confidence but had large negative $\mathbb{LO}$ values. Sampling and adding noise vectors to these images and re-predicting their labels confirm that the original predictions are not stable, as the prediction switches to a different label majority of the time.

| $r$ | $r = 0.5$ | | $r = 1.0$ | | $r = 2.0$ | |
|---|---|---|---|---|---|---|
| $\mathbb{LO}$ | Top | Bottom | Top | Bottom | Top | Bottom |
| MNIST | 0.02% | 5.1% | 2.0% | 9.9% | 3.4% | 19.4% |
| CIFAR-10 | 1.8% | 2.2% | 2.0% | 22.1% | 2.0% | 59.9% |
| CRC | 0.07% | 12.3% | 0.18% | 18.9% | 0.05% | 27.6% |
| SVHN | 0% | 1.6% | 0% | 6.1% | 0% | 21.6% |

Table 1: Label flip percentages for all datasets. 1K random test objects were selected from each dataset. The test objects whose label were predicted with at least 0.8 probability were subjected to random noise whose length was $r$. For these confident objects, we computed what percentage of those samples changed their predicted labels under noise. The group that had the smallest (i.e, largest negative) $\mathbb{LO}$ value had higher label flip percentages.

In our next analysis, we group the test objects into two: those who have at least one noisy version (among 10K samples) whose label is different from the original test object versus those whose *all* noisy versions' labels agree with the original test object's label. We present the distribution of $\mathbb{LO}$ values as box plots and the p-values for the unpaired t-tests comparing "flipped at least once" to "never flip" for each $r$ in Figure 4. As the results show, $\mathbb{LO}$ is significantly smaller (more negative) for the "flipped at least once" group.

### 4.3 $|\mathbb{LO}|$ and Corrupted Data

In this section, we analyze two benchmark robustness datasets: MNIST-C by Mu & Gilmer (2019) and CIFAR-10-C by Hendrycks & Dietterich (2019). The MNIST-C dataset contains MNIST images corrupted by 15 different corruption methods such as *guassian_blur*, *scatter*, and etc. Each original test image of the MNIST dataset is corrupted separately using one of these 15 corruption methods, which leads to 15 groups of corrupted images. The CIFAR-10-C dataset has 19 different corruption methods and each corruption has 5 levels of corruption that can be added to each image.

We expect the original images to have more stable predictions compared to their corrupted versions. Figure 5a compares the $|\mathbb{LO}|$ on all corrupted sets and the original images for the MNIST-C dataset, and figure 5b on all corrupted image with corruption level 3 and the original images for the CIFAR-10 dataset. (Appendix includes the comparison with other corruption levels). The result for the original images is shown as a dotted line because it does not depend on the corruption method. The results show that the corrupted images always have a higher $|\mathbb{LO}|$ than original images from MNIST, and for majority of corrupted images for CIFAR-10.

### 4.4 $|\mathbb{LO}|$ and Model robustness

In this section, we study the connection between the robustness of the model and values of $|\mathbb{LO}|$ as discussed in Section 3.1.3. We compare a "standard" model that is trained using traditional learning methods with models that are trained using adversarial training. To be clear, our primary objective is not to conduct a comprehensive evaluation of adversarial training methods to enhance the robustness of the model. However, we utilize the adversarial training process along with $|\mathbb{LO}|$ to evaluate model robustness from a random

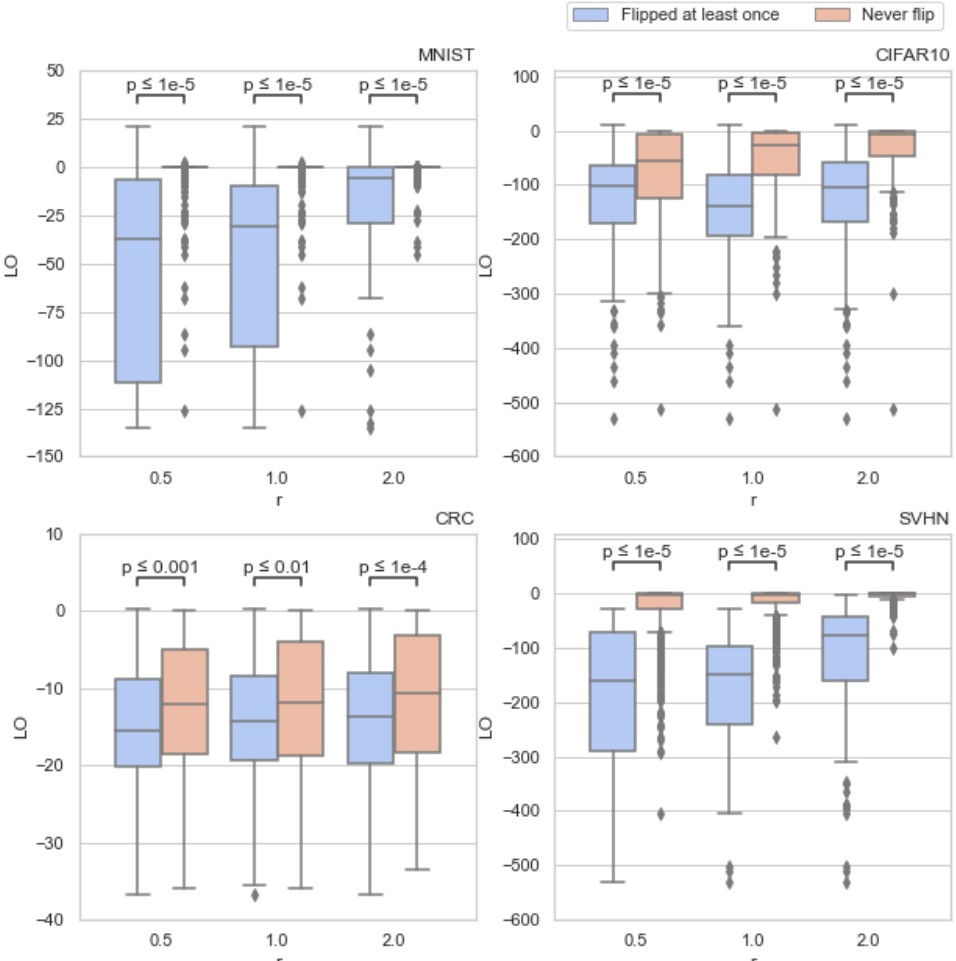

Figure 4: Average $\mathbb{LO}$ within two groups. *Flipped at least once* contains the test objects who has at least one noisy version whose label has changed. The *Never flip* contains the objects whose noisy versions all agree with the test object's label. Noisy versions are created by adding random noise with length $r$ to original object.

noise perspective. For adversarial training of the model, we adopt the strategy from Tramèr et al. (2018); Goodfellow et al. (2015). Table 2 summarizes the accuracies and the $L_2$ norm of the weights of these models. As the table shows, these models have relatively comparable accuracies, except for CIFAR-10 dataset where the accuracy drops 10% for adversarial training. The adversarial model for MNIST has a significantly lower $L_2$ norm (suggesting a simpler model), whereas the $L_2$ norms are comparable for the other datasets (suggesting similar model complexity).

We next compute $|\mathbb{LO}|$ for each model for each test data point. We present the boxplots, in Figure 6, of these $|\mathbb{LO}|$ values and the p-values of the t-test comparing the adversarially-trained model and standard-trained model. We observed that the predictions trained by adversarial training have statistically significantly lower $|\mathbb{LO}|$ values for all datasets. Morever, as expected, $|\mathbb{LO}|$ is smaller if the input dimension is smaller (the equation 2), hence the $|\mathbb{LO}|$ values for MNIST are smaller, in absolute terms, than the other three datasets. This leaves a smaller room for objects to differ in $|\mathbb{LO}|$ values. However, the reported p-value indicates a statistically significant difference in the $|\mathbb{LO}|$ values for adversarial-trained versus standard-trained models.

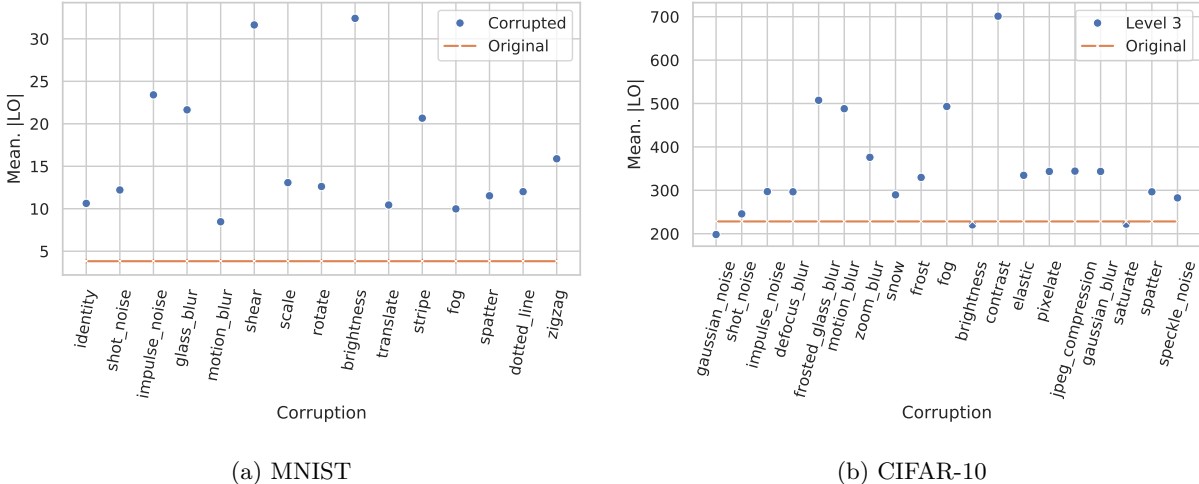

(a) MNIST

(b) CIFAR-10

Figure 5: Comparing average $|\mathbb{LO}|$ for the original images (dotted line) versus the corrupted groups of images for the MNIST-C and the CIFAR-10-C datasets. $|\mathbb{LO}|$ is always higher for all the corrupted images from MNIST-C, and higher for most of the corrupted images from CIFAR-10-C. Results for other levels of corruption for CIFAR-10-C are included in the appendix.

| Dataset | MNIST | CIFAR-10 | CRC | SVHN |
|---|---|---|---|---|
| Dimension | 28×28×1 | 32×32×3 | 27×27×3 | 32×32×3 |
| Accuracy (Standard Train) | .98 | .85 | .74 | .92 |
| Accuracy (Adversarial Train) | .99 | .75 | .71 | .89 |
| $L_2$ of weights (Standard Train) | 1933 | 1190 | 1130 | 1041 |
| $L_2$ of weights (Adversarial Train) | 1492 | 1006 | 1335 | 1228 |

Table 2: Summary of model performance from standard training and adversarial training. Adversarial training results lower or similar training accuracy as expected compared to the standard training method. $L_2$ norm is smaller for the adversarial model for MNIST (suggesting a simpler model), and comparable for other models (suggesting comparable complexity). Note that the models that correspond to "standard" training and "adversarial" training have the same model architecture.

## 5 Limitations and Future work

The exact computation of $\mathbb{LO}$ requires the second derivatives $\frac{\partial^2 f}{\partial x_i{}^2}$. These derivatives can be derived analytically for a given neural network structure and can be computed efficiently based on this analytical derivation. However, most packages including Tensorflow, do not support the computation of the second derivate of $f$ with respect to input $x$. Some instead provide the computation of the Hessian matrix, which is computationally expensive, and is an overkill for computing $\mathbb{LO}$, as $\mathbb{LO}$ needs only the diagonals of the Hessian matrix, and not the full matrix. Thus, using $\mathbb{LO}$ for robustness is difficult for many of the state-of-the-art models.

In this paper, we studied the mean change in the predicted probability with respect to perturbations, $\mathbb{E}[\Delta f(x)]$. We showed that for high confidence predictions, a large negative $\mathbb{LO}$ can indicate a label flip under small perturbations. Though this quantity can also be used to study the stability of the probability distribution itself, it measures only the mean change. A promising future direction is to derive the variance of the probability change, with respect to perturbations: $\mathbb{E}[(\Delta f(x) - \mathbb{E}(\Delta f(x))^2]$.

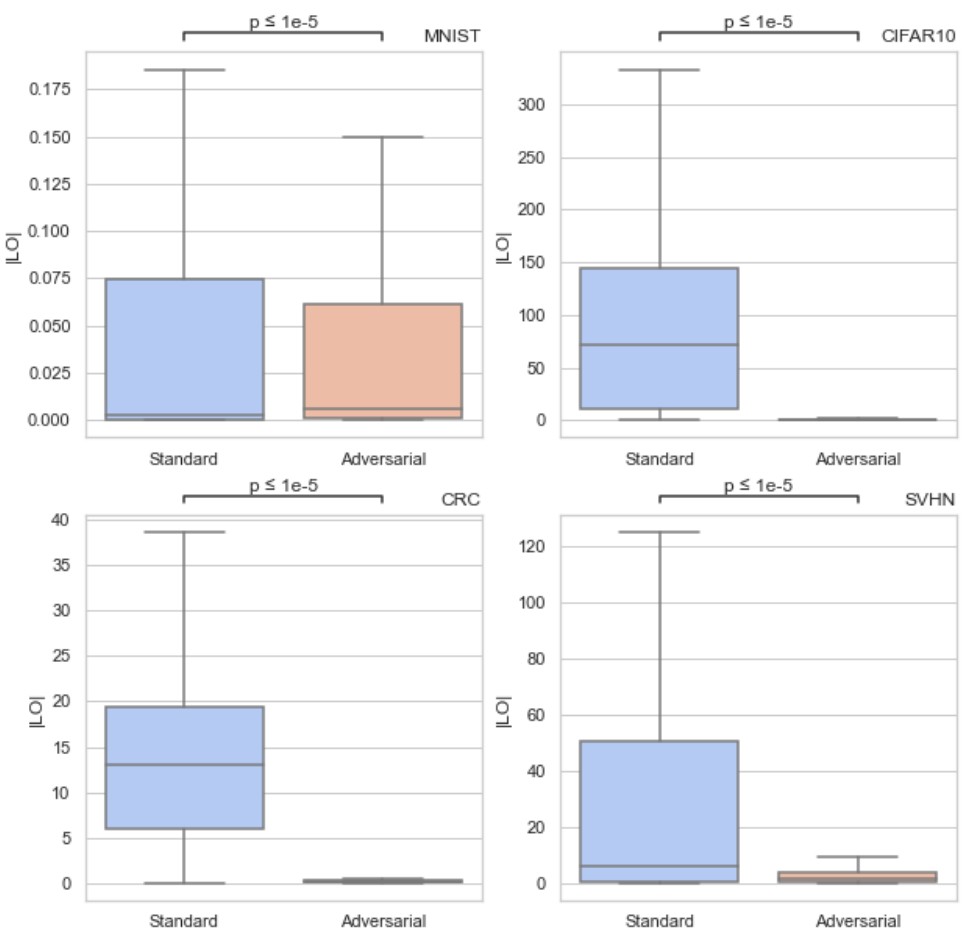

Figure 6: The model robustness and $|\mathbb{LO}|$ for each dataset. The p-values show that the models that are trained using adversarial training have statistically significantly lower $|\mathbb{LO}|$ values compared to the same architectures that are trained using standard procedures.

## 6 Conclusions

We investigated the robustness of the probability of the predicted class for a differentiable model. We derived using the Taylor expansion and Divergence theorem that the expected change in the probability of the predicted class, with respect to random perturbations around the input, is a multiple of the Laplace operator at the input. We conducted empirical analyses on four image classification datasets. The first empirical study showed that we were able to identify objects whose labels were predicted with high confidence but yet were still unstable under random noise. The second empirical study on two robustness benchmark datasets showed that the absolute value of the Laplace operator was higher for corrupted images than the original ones. Lastly, the experiments demonstrated that $|\mathbb{LO}|$ can distinguish with statistical significance a standard model from a model that is trained using adversarial training, and hence can be used as a measure of overall model robustness.

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

## A   Appendix

### A.1   Lemmas and Proofs

**Lemma 4.** *Define* $\Delta f_{c,S'}(\overrightarrow{x_0}) = f(\overrightarrow{x_0} + \Delta \overrightarrow{x}) - f_c(\overrightarrow{x_0})$, *where* $\{\overrightarrow{x_0}, \Delta \overrightarrow{x}\} \in R^n$, $\Gamma(.)$ *is the Gamma Function,* $r \in R^+$, $S' \equiv ||\overrightarrow{x} - \overrightarrow{x_0}||_2 = r$. *Then the expected Change around point* $\overrightarrow{x_0}$:

$$E(Change) = \frac{\Gamma(\frac{n}{2})}{2\pi^{\frac{n}{2}}} r^{1-n} \int_{S'} \Delta f_{c,S'}(\overrightarrow{x_0}) dS' \tag{8}$$

*Proof.* To prove the Lemma 1, we first define a *n dimensional ball* with radius $r$ in the space and centered at $\overrightarrow{x_0}$. The $S'$ is the $n-1$ *dimensional sphere* of the ball such that any point $\overrightarrow{x}$, $\overrightarrow{x} = \overrightarrow{x_0} + \Delta \overrightarrow{x}$, with the Euclidean distance $r$ to the point $\overrightarrow{x_0}$ is on the sphere $S'$, $\overrightarrow{x} \in S'$. In other words, with attack vector which has length $r$, $|\Delta \overrightarrow{x}| = r$, every attack sample point $\overrightarrow{x}$ around un-attacked sample $\overrightarrow{x_0}$ is on the surface $S'$. Then define a infinitely small region $A$ on the surface $S'$ such that $\overrightarrow{x} \in A$. The probability density function $p(\overrightarrow{x})$ on the surface $S'$ is the relative likelihood for a point on sphere $S'$ and also in the region of $A$. Formally, $Prob(\overrightarrow{x} \in A) = \int_A p(\overrightarrow{x}) d(\overrightarrow{x})$. Roughly speaking, $p(\overrightarrow{x})$ measures the likelihood of a point $\overrightarrow{x}$ to be selected on the surface $S'$ and of course for any particular point $\overrightarrow{x}$, the measure is 0, that is $p(\overrightarrow{x})$ for a single point is 0 since it is a probability density function. There is a *Change* respect to each possible attack sample $\overrightarrow{x}$. So the expectation of *Change* is the integral over each *Change* on its point $\overrightarrow{x}$ times the likelihood of the point $\overrightarrow{x}$. Formally, this integral in a surface integral over $S'$, so:

$$E(\Delta f_{c,S'}(\overrightarrow{x_0})) = E(f_c(\overrightarrow{x_0} + \Delta \overrightarrow{x}) - f_c(\overrightarrow{x_0}))$$

$$= \int_{S'} \Delta f_{c,S'}(\overrightarrow{x_0}) p(\overrightarrow{x}) d(\overrightarrow{x})$$

$$\equiv \int_{S'} \Delta f_{c,S'}(\overrightarrow{x_0}) p(\overrightarrow{x}) dS' \tag{9}$$

$p(\overrightarrow{x})$ is the probability density function of a point on a $(n-1)$ *Dimension* sphere $S'$, so the integral of $p(\overrightarrow{x})$ over all $\overrightarrow{x}$ is 1. That is:

$$\int_{S'} p(\overrightarrow{x}) dS' = 1 \tag{10}$$

Since sphere $S'$ is a surface of a *ball*, so every point on sphere $S'$ has equal likelihood to be selected, then probability density function $p(\overrightarrow{x})$ is a uniform probability distribution, which means $p(\overrightarrow{x})$ is a constant for a fixed dimension $n$ , then:

$$p(\overrightarrow{x}) \int_{S'} 1 dS' = 1 \tag{11}$$

For any given $n\ dimensional\ ball$ with radius $r$, the area of the surface $S$ is $\int_S 1 dS = \frac{2\pi^{\frac{n}{2}}}{\Gamma(\frac{n}{2})} r^{n-1}$, where $\Gamma(.)$ is the Gamma function, so:

$$p(\overrightarrow{x}) = \frac{\Gamma(\frac{n}{2})}{2\pi^{\frac{n}{2}}} r^{1-n} \tag{12}$$

and:

$$Eq.(9) = \frac{\Gamma(\frac{n}{2})}{2\pi^{\frac{n}{2}}} r^{1-n} \int_{S'} \Delta f_{c,S'}(\overrightarrow{x_0}) dS' \tag{13}$$

It completes the proof of Lemma 1. □

**Lemma 5.** *$S$ is a surface of the $n$ dimensional ball $V$ centered at $\overrightarrow{0}$ with radius $r$. $\overrightarrow{x} = (x_1, \ldots, x_n) \in S$, then: $\int_S x_i dS = 0$, for $1 \leqslant i \leqslant n$.*

*Proof.* To prove Lemma 2, we first construct a vector filed $F^{(i)}$ for each $i \in [1, n]$ with value is 1 for $i^{th}$ dimension and 0 for others.

$$F^{(i)} = (0, \ldots, \underbrace{1}_{i^{th}}, \ldots, 0) \tag{14}$$

Then, the unit normal vector directed outward from $V$ is:

$$\overrightarrow{n} = (\frac{x_1}{r}, \frac{x_2}{r}, \ldots, \frac{x_n}{r}) \tag{15}$$

and:

$$F^{(i)} \cdot \overrightarrow{n} = \frac{x_i}{r} \tag{16}$$

Then:

$$\int_S x_i dS = r \cdot \int_S F^{(i)} \cdot \overrightarrow{n} dS \tag{17}$$

The divergence of the vector filed is 0:

$$\nabla \cdot F^{(i)} = \sum_{j=1}^{n} \frac{\partial F_j^{(i)}}{\partial x_j} = 0 \tag{18}$$

Clearly, $F^{(i)}$ is a vector field whose component functions have a continuous partial derivatives in $V$, so by the statement of The *Divergence Theorem*, we have:

$$\int_S F^{(i)} \cdot \overrightarrow{n} dS = \iint_V (\nabla \cdot F^{(i)}) dV = \iint_V 0 dV = 0 \tag{19}$$

This completes the proof of Lemma 2 □

**Lemma 6.** *$S$ is a surface of the $n$ dimensional ball $V$ centered at $\overrightarrow{0}$ with radius $r$. $\overrightarrow{x} = (x_1, \ldots, x_n) \in S$, then: $\int_S x_i^2 dS = \frac{\pi^{\frac{n}{2}}}{\Gamma(\frac{n}{2}+1)} r^{n+1}$, for $1 \leqslant i \leqslant n$.*

*Proof.* To prove Lemma 3, we use the similar idea from above, but construct a vector filed $F^{(i)}$ for each $i \in [1, n]$ with value is $x_i$ for $i^{th}$ dimension and 0 for others.

$$F^{(i)} = (0, \ldots, \underbrace{x_i}_{i^{th}}, \ldots, 0) \tag{20}$$

The divergence of the vector filed is 1:

$$\nabla \cdot F^{(i)} = \sum_{j=1}^{n} \frac{\partial F_j^{(i)}}{\partial x_j} = 1 \tag{21}$$

so by the *Divergence Theorem*:

$$\int_S x_i{}^2 dS = r \cdot \int_S F^{(i)} \cdot \overrightarrow{n} \, dS = r \cdot \int_V 1 dV \tag{22}$$

$\iint_V 1 dV$ is the volume of the *ball V*, and for any *n dimensional ball*, the volume Gipple (2014) is: $\frac{\pi^{\frac{n}{2}}}{\Gamma(\frac{n}{2}+1)} r^n$:

$$\iint_V 1 dV = \frac{\pi^{\frac{n}{2}}}{\Gamma(\frac{n}{2}+1)} r^n \tag{23}$$

So:

$$\int_S x_i{}^2 dS = \frac{\pi^{\frac{n}{2}}}{\Gamma(\frac{n}{2}+1)} r^{n+1} \tag{24}$$

$\square$

**Lemma 7.** *S is a surface of the n dimensional ball V centered at $\overrightarrow{0}$ with radius r. $\overrightarrow{x} = (x_1, \ldots, x_n) \in S$, then: $\int_S x_i x_j dS = 0$, for $1 \leqslant i \leqslant n$, $1 \leqslant j \leqslant n$, and $i \neq j$.*

*Proof.* By the similar idea, we construct a vector filed $F^{(i)}$ for each pairs of $i \neq j \in [1, n]$ with value $\beta x_j$ for $i^{th}$ dimension, $(1 - \beta)x_i$ for $j^{th}$ dimension, and 0 for others, where $\beta \in [0, 1]$.

$$F^{(i,j)} = (0, \ldots, \underbrace{\beta x_j}_{i^{th}}, \ldots, \underbrace{(1-\beta)x_i}_{j^{th}}, \ldots, 0) \tag{25}$$

Then, the divergence of this vector filed is 0:

$$\nabla \cdot F^{(i,j)} = \sum_{j=1}^{n} \frac{\partial F_j^{(i)}}{\partial x_j} = 0 \tag{26}$$

This completes the proof by using the *Divergence Theorem* again. $\square$

### A.2 Additional figures

### A.3 Model details

The MNIST dataset is split into train and test as follows: 60K images are on the training set and the test set has 10K images. We adopt the LeNet architecture from LeCun et al. (1998) and train on MNIST with normalized pixel values. Specifically, following the settings in LeCun et al. (1998), we use the hyperbolic tangent function as activation function for all convolution layers, and then linear activations are used for the dense layers. We utilize all samples in the MNIST training set to train with 20 epochs, batch size 500, and Adam Kingma & Ba (2015) optimizer with learning rate 0.01. The accuracy of this model on the test set is 98%.

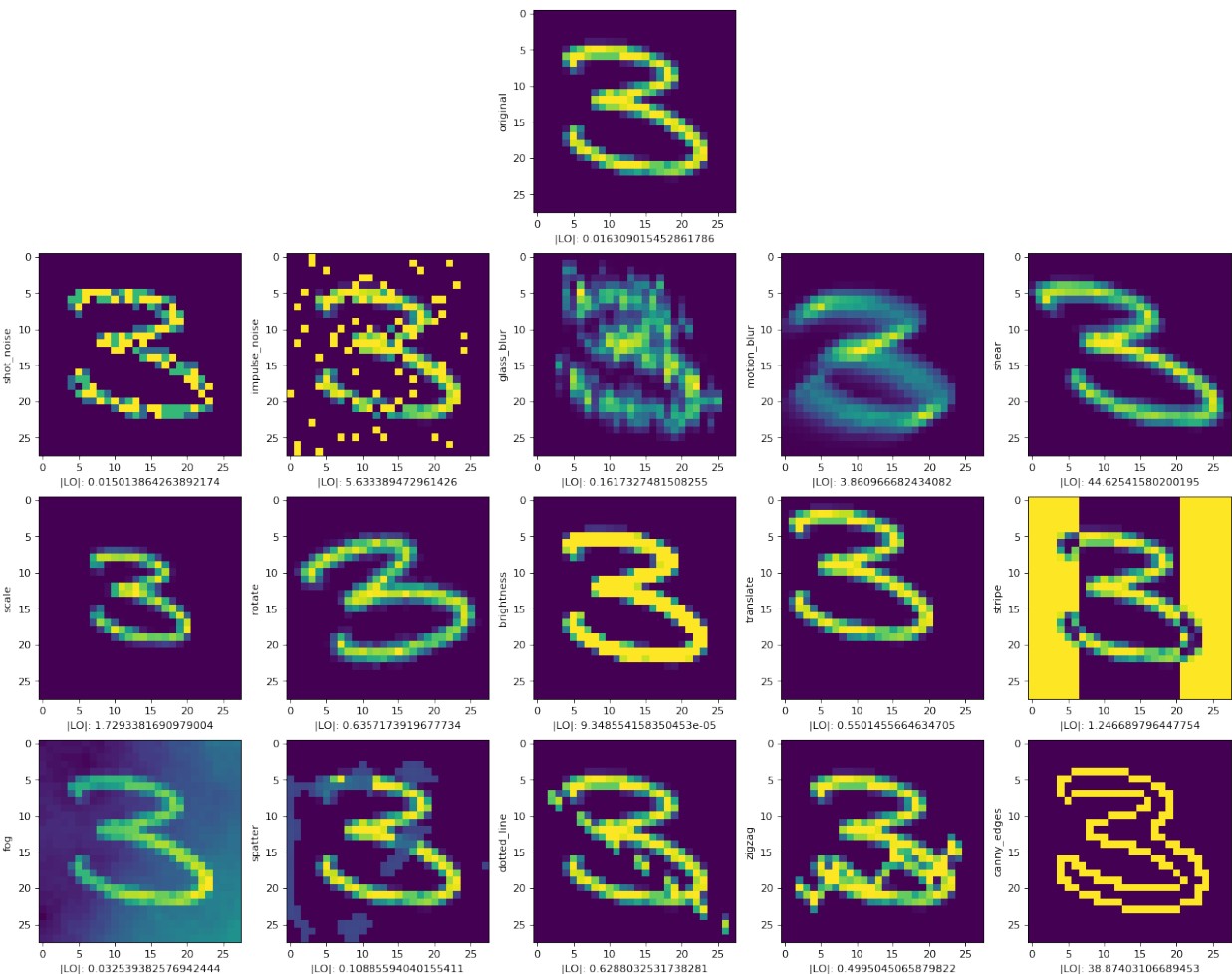

Figure 7: Clean and Corrupted Images with $|\mathbb{LO}|$

The CIFAR-10 dataset is split into train, validation, and test as follows: 45K images are on the training set, 5K images are on the validation set, and the test set has 10K images Abadi et al. (2015). We create a ResNet He et al. (2016) model for this dataset. We create 3 Res-blocks to construct the ResNet model. We initialize all weights following He et al. (2015a) and utilize the PRelu He et al. (2015b) activation function. We use data augmentation before passing the training data into the model, where we randomly horizontally flip, and shift both height and width with maximum 12.5% range. Furthermore, we use a batch size of 128, regularization constant of 0.0001, learning rate of 0.001, and SGD with momentum of 0.9, and we optimize the epoch number using the validation accuracy. The final model achieves top-1 test accuracy of 85%.

For SVHN dataset, we adapt the same model structure and training parameters of CIFAR-10 to SVHN dataset Netzer et al. (2011). We hold original test dataset (26032 objects) for testing, and randomly reserve 5K objects from training sets as validation set. The accuracy on test dataset is 92%.

Sirinukunwattana et al. (2016) introduced the histology images colorectal cancer dataset (CRC), which contains 100 H&E stained colorectal adenocarcinomas images where each image contains many cells. The cells in the stained images are labeled as: *Epithelial*, *Inflammatory*, *Fibroblast*, or *Miscellaneous*, and the location of the center of each labeled cell is provided in the data. We extract a 27x27x3 image for each cell at the locations provided in the data. The total number samples of each class are 7,057, 6,278, 5,130, and 1,842. We split the dataset into 70% train, 15% validation, and 15% test. We used the same ResNet architecture that we used for CIFAR-10 and SVHN, but changed the input and output size as needed. The model achieves a top-1 test accuracy of 74%.

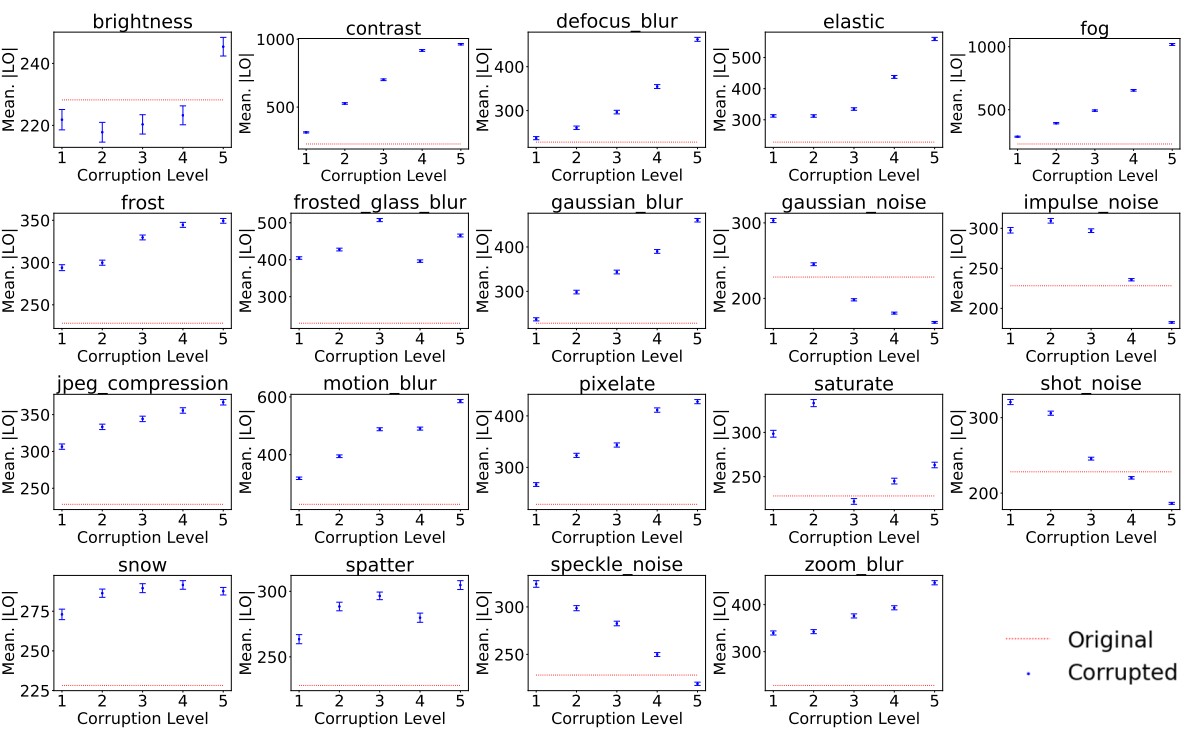

Figure 8: Comparing average $|\mathbb{LO}|$ for the original images (dotted line) versus the corrupted groups of images for the CIFAR-10-C dataset. $|\mathbb{LO}|$ is almost (83 out of 95) always higher for the corrupted images. $|\mathbb{LO}|$ is positively correlated with the corruption level for most corruption methods.

