# OpenReview forum: "The Analysis of the Expected Change in the Classification Probability of the Predicted Label"
_TMLR — Accepted by TMLR_

### Review · Reviewer_sGRe · 2023-06-07

**Summary Of Contributions:**

The authors prove that the expected change in a function's classification probability for random perturbations that lie within a fixed radius around a data point can be estimated using the Laplace operator of the function.  They then conduct experiments on a variety of datasets (MNIST, CIFAR-10, SVHN, CRC) and demonstrate that the Laplace operator is correlated with the rate at which the classifier flips the predicted label for noisy images and model robustness.

**Audience:**

Yes

**Claims And Evidence:**

Yes

**Requested Changes:**

See the point under Weaknesses, addressing this point would strengthen the work in my view.

**Strengths And Weaknesses:**

Strengths:
- paper presentation is clear
- problem studied is interesting and well-motivated
- theoretical contributions are interesting and findings are backed by experimental results
- good experimental scope across datasets

Weaknesses:
- In Section 3, the authors list 3 practical applications (robustness of predicted label, robustness of probability distribution, robustness of model) of the Laplace operator.  I think the experimental results provided generally demonstrate the first and third applications, but it doesn't seem like there are experimental results that demonstrate the second applications.  Since the authors mention model calibration and uncertainty quantification in the related works section, I wonder if experiments could be used to demonstrate that more calibrated models have Laplace operator closer to 0 than uncalibrated models.

---

> ### Author Response · Authors · 2023-07-07
>
> Thank you for the comments and the feedback.
>
>
> While we did not measure the robustness of the full probability spectrum, Table 1 presents results for high-confidence objects (over the probability of 0.8) and shows that even though they are predicted with high confidence, not all of them are robust to noise. $\mathbb{LO}$ correlates well with the ones whose label flips (i.e., the probability distribution changes so much that the label flips). In other words, experiments show that the smaller (i.e., larger negative) of $\mathbb{LO}$ is, the less robust the probability distribution at the local point $x$ is.

---

### Review · Reviewer_b491 · 2023-06-22

**Summary Of Contributions:**

This paper proposes a new estimation of the expected change in the probability distribution with respect to the small perturbations, offering a potential tool for assessing the robustness of the model prediction. Specifically, the authors analytically derive that the Laplacian operator becomes such an estimation. With this estimation, they demonstrate its effectiveness across three different applications on four different computer vision benchmarks: 1) distinguishing not-robust prediction with high confidence, 2) detecting corrupted images, and 3) measuring the adversarial robustness of neural networks.

**Audience:**

Yes

**Broader Impact Concerns:**

There are no broader impact concerns.

**Claims And Evidence:**

Yes

**Requested Changes:**

Please address the concerns above. Specifically,

1. Clarification of the contribution and comparison with previous works
2. Comparison with baseline methods
3. Experiments with larger benchmark

**Strengths And Weaknesses:**

**Strengths.**

- **Well-motivated problem.** Measuring the robustness of model prediction is a practically important direction for its reliable deployment to the real world.
- **Intuitive approach.** The proposed estimation via the Laplacian operator is intuitive and well-grounded by the provided derivation.

**Weaknesses.**

- **Novelty.** The second derivation (or Hessian) has been widely studied in the literature on model robustness [1,2,3,4]. Although the authors propose new applications with this, it’s hard to find a technical novelty compare to the existing works.
- **Lack of baseline**. The authors do not compare with the naive baselines. For example, instead of Laplacian Operator, one can use model confidence or entropy for selecting Top and Bottom samples in Table 1. Also, Figure 4 can be drawn by adopting other measurements as y-axis. Hence, the authors should provide such baseline results and validate that the proposed estimation is better than other choices.
- **Limited experimental demonstration**. The authors only adopt the small models (LeNet or 3-layer ResNet) and benchmarks (MNIST, SVHN, CNC, and CIFAR-10) for the experiments. However, the proposed scheme can be easily applied to the larger scale of benchmarks (e.g., ImageNet, CIFAR100) with state-of-the-art models (e.g., pre-trained ResNet or ViT). If the authors can show that the proposed method also well performs in those scenarios, it will be really interesting and valuable.

[1] Ros and Doshi., Improving the Adversarial Robustness and Interpretability of Deep Neural Networks by Regularizing Their Input Gradients., AAAI 2018 \\\
[2] Yao et al., Hessian-based Analysis of Large Batch Training and Robustness to Adversaries., NeurIPS 2018 \\\
[3] Singla and Feizi., Second-Order Provable Defenses against Adversarial Attacks., ICML 2020 \\\
[4] Tsiligkaridis and Roberts., Second Order Optimization for Adversarial Robustness and Interpretability., arXiv \\

---

> ### Author Response · Authors · 2023-07-07
>
> Thank you for the comments and the feedback.
>
> Re: novelty: first and most importantly, even though we use $\mathbb{LO}$ in our experiments, our paper’s main contribution is not “the experimental evaluation of $\mathbb{LO}$ for robustness.” Rather, our objective in the paper is to derive analytically the expected change in probability distribution due to random perturbations. This “analytical derivation of expected probability change due to random noise” is not presented in any previous work to our knowledge. The derivation ended up being proportional to the sum of the Hessian's diagonal, which is a welcome result given that the Hessian is extensively studied. Hence, our main contribution is this derivation.
>
> Re: baseline: We already use the confidence-based approach in Table 1. The prediction from the model for a given object is intuitively robust when the model predicts a high degree of confidence for the object. We emphasize that, despite the fact that the model provides high-confidence predictions for some objects, the labels can still flip under random noise, our method highlights these objects with a high degree of confidence.
>
> Comparing our paper to those mentioned in the review, we find the following:
> The Ros and Doshi paper ([1]) presents a first-order method that does capture the local curvature of a given function. Moreover, the measurement for local robustness was not provided in the paper, but the gradient was regularized during training. Our method derives expected probability change, makes use of the local curvature information, and provides a measurement of the local probability robustness.
> The paper by Yao et al. ([2]) uses the Hissen spectrum, which theoretically requires the computation of the entire Hissen matrix ($O(n^2)$), and the spectrum itself does not approximate the expected probability change. Our method only requires $O(n)$ time to calculate the diagonal of the Hessian matrix, as we do not need the full Hessian matrix. As in the paper by Tsiligkaridis and Roberts. ([4]), the target function for both papers ([2] and [4]) is a loss function, which means they focus on researching the loss function's curvature during training. We focus on the robustness of an already trained model, such as a convolutional neural network that has already been trained on a training dataset that is not necessarily available anymore. Both [2] and [4] investigate $L(x, y)$, whereas we investigate $f(x)$.
>
> Singla and Feizi's ([3]) paper focuses on locating and enhancing the certified robustness of input using curvature information. For instance, what is the minimum radius that, when applied to the input, will cause the prediction for a given input to change? The certification ensures that no label will be flipped within the specified distance. Our approach is not intended to provide certification. We are measuring the expected change in probability based on a radius and all possible perturbations.
>
> Re: benchmarks
>
> We use the same classical datasets (MNIST, CIFAR-10, SHVN) as [1], [2], [3], and [4].  Moreover,  [1] only used MNIST and SHVN, [3] only MNIST, [4] only CIFAR-10 and SHVN, and [2] both CIFAR-10 and CIFAR-100. Note that the input dimensions for CIFAR-10 and CIFAR-100 are identical; however, CIFAR-10 contains only 10 classes, whereas CIFAR-100 includes 100 classes.
> To compare the model size with mentioned papers, all of the experiments in the [1] paper used a model of comparable size to LeNet. Experiments in the [3] paper used only feedforward neural networks. In our experiments, we used both LeNet and ResNet. [2] used models of comparable size to ours for experiments with CIFAR-10 and MNIST, but a more complex ResNet for CIFAR-100 experiments.
> Regarding larger models, the computation for the $\mathbb{LO}$ is costly due to the implementation of current deep-learning packages, where the full Hessian is computed; our method requires only the diagonal of Hessian, which theoretically requires only $O(n)$ computational cost as opposed to the full Hissen, which requires $O(n^2)$ computational cost. This was acknowledged in the paper's limitations section.

---

### Review · Reviewer_jsEK · 2023-06-24

**Summary Of Contributions:**

This paper proposes a novel framework for assessing the change in predicted label probabilities for an object in response to small perturbations. The authors first develop an analytical estimate for expected probability alterations in relation to input noise. Following this, they substantiate their findings with three empirical studies. The first experiment demonstrates that their proposed metric can differentiate between robust and non-robust label predictions, even when these predictions are made with high confidence. The second study reveals that their measure typically indicates greater robustness for predictions made on corrupted images, compared to those made on their original counterparts. Lastly, the study finds that their proposed robustness measure tends to be lower for models that have undergone adversarial training. In summary, this paper provides an innovative analytical approach, backed by empirical studies, to better understand and measure the impact of small perturbations on label prediction probabilities.

**Audience:**

Yes

**Broader Impact Concerns:**

There is no broader impact concern regarding this paper.

**Claims And Evidence:**

Yes

**Requested Changes:**

The paper's experimental justification is somewhat limited due to the lack of incorporation of advanced adversarial training methods as baselines. This omission could potentially undermine the full validation of the proposed method and its efficiency, thus necessitating more comprehensive experiments.

**Strengths And Weaknesses:**

Strengths:

This paper formalizes a new idea to perform adversarial machine learning.

This paper is easy to follow, and the main contribution is clearly stated.

Weaknesses:

This paper presents a novel approach to evaluating how minor perturbations can impact the predicted label probabilities of an object. Initially, the authors present an analytical estimate of anticipated probability variations tied to input noise. Subsequently, they conduct three empirical studies to support their theoretical findings. The first study illustrates the ability of their proposed metric to distinguish robust from non-robust label predictions, even when predictions exhibit high confidence. In the second study, predictions on corrupted images consistently show a higher measure of robustness in comparison to those on original images. Finally, models subjected to adversarial training approaches exhibit lower values of the proposed measure, as demonstrated in the last study. However, the paper's experimental justification is somewhat limited due to the lack of incorporation of advanced adversarial training methods as baselines. This omission could potentially undermine the full validation of the proposed method and its efficiency, thus necessitating more comprehensive experiments.

---

> ### Author Response · Authors · 2023-07-07
>
> Thank you for the comments and the feedback.
>
> The objective of this paper is to evaluate the robustness of the predicted probability to **random** noise. In particular, we analyze and quantify the expected probability change due to all possible random noise on an $l_2$ ball surface. Random noise can exist in real-world applications due to noisy and low-precision sensors, such as low resolution. Even though adversarial training experiments are relevant for the paper, they are only relevant to the extent that they make the model more robust to random noise. Because many adversarial attacks, such as the one-pixel attack, are targeted and not random, our method is not applicable to those attacks. In other words, the adversarial training process might produce robust models (to attacks but also to noise) compared to adversarial training methods. By computing the local $\mathbb{LO}$ for each sample, we demonstrate that the $|\mathbb{LO}|$ is indeed lower for models that are trained using adversarial training methods. However, our primary objective is not to conduct a comprehensive evaluation of adversarial training methods to enhance the robustness of the model. Having a smaller $|\mathbb{LO}|$ signifies robustness to random noise but it does not signify robustness to targeted attacks. We will clarify this in the paper.

---

### Decision · Action_Editors · 2023-08-18

**Recommendation:** Accept with minor revision

**Comment:**

This paper proposes an analytical estimate of the expected change in the probability distribution for random perturbations on an $l_2$ ball surface. The idea of using the second derivation (or Hessian's diagonal) is not surprising, but the potential to evaluate the robustness of the classifier is interesting. The reviewers had a number of concerns about novelty, contributions, and experiments, which were mostly resolved through the discussion process. Overall, this is an interesting and useful paper and I recommend its acceptance.

---

- As in the author's comments to Reviewer jsEK, related clarification should be clearly demonstrated in the paper to avoid an overclaim.
- As Reviewer b491 also requested, I highly recommend conducting experiments with a larger benchmark. I understand that computing the Hessian diagonal will be a bottleneck, but recent advances in deep learning frameworks can help accelerate this computation, e.g., a library for PyTorch - https://docs.backpack.pt/en/master/extensions.html#backpack.extensions.BatchDiagHessian.

**Audience:**

Yes, the paper is interesting to researchers working on robust AI.

**Claims And Evidence:**

Yes, the paper provides an analytical derivation.